# Tree Species Classification Based on Upper Crown Morphology Captured by Uncrewed Aircraft System Lidar Data

Robert J. McGaughey [1,*], Ally Kruper [2,3], Courtney R. Bobsin [2,3] and Bernard T. Bormann [2,3]

1   Pacific Northwest Research Station, USDA Forest Service, Martinsville, IN 46151-9718, USA
2   School of Environmental and Forest Sciences, University of Washington, Seattle, WA 98195, USA;
    akrup99@uw.edu (A.K.); cbobsin@uw.edu (C.R.B.); bormann@uw.edu (B.T.B.)
3   Olympic Natural Resources Center, School of Environmental and Forest Sciences,
    University of Washington, Forks, WA 98331, USA
*   Correspondence: robert.mcgaughey@usda.gov

**Abstract:** The application of lidar data to assist with forest inventory is common around the world. However, the determination of tree species is still somewhat elusive. Lidar data collected using UAS (uncrewed aircraft systems) platforms offer high density point cloud data for areas from a few to several hundred hectares. General point cloud metrics computed using these data captured differences in the crown structure that proved useful for species classification. For our study, we manually adjusted plot and tree locations to align field trees and UAS lidar point data and computed common descriptive metrics using a small cylindrical sample of points designed to capture the top three meters and leader of each tree. These metrics were used to train a random forest classifier to differentiate between two conifer species, Douglas fir and western hemlock, common in the Pacific Northwest region of the United States. Our UAS lidar data had a single swath pulse density of 90 pulses/m$^2$ and an aggregate pulse density of 556 pulses/m$^2$. We trained classification models using both height and intensity metrics, height metrics alone, intensity metrics alone, and a small subset of five metrics, and achieved overall accuracies of 91.8%, 88.7%, 78.6%, and 91.5%, respectively. Overall, we showed that UAS lidar data captured morphological differences between the upper crowns of our two target species and produced a classification model that could be applied over large areas.

**Keywords:** UAS; lidar; tree species; random forest; classification; crown morphology; Pacific Northwest

## 1. Introduction

Forest inventory is the systematic collection of data and forest information for assessment or analysis. A forest stand inventory uses tree measurements collected at a network of sample locations to produce summaries for variables of interest such as species distribution, basal area, volume, stem density and biomass. Given the cost of field data collection, especially in rugged terrain and areas with limited road access, foresters have looked to remote sensing methods to augment or replace traditional field-based inventories. Beginning in the mid-1980s, laser profiling and scanning systems were used to quantify tree and stand characteristics [1–3]. Næsset [4,5] expanded on this work and showed that lidar had the potential to improve stand volume and height estimates compared to traditional methods involving aerial photointerpretation. Further work by numerous researchers has brought us to a point where lidar-assisted inventories are now commonplace [6–9].

The use of lidar to provide measurements of tree and stand attributes is well documented in the literature. However, the use of lidar data to provide species information is generally limited to applications involving a few species or small areas. In their review of species classification using lidar, Michalowska and Rapinski [10] found that most lidar studies that focused on species classification involved six or fewer species. Overall, the accuracies reported in these studies ranged from 97 to 47%, with accuracy generally decreasing as the number of species increased. The general approach when using lidar data

to assist with inventory or species classification is to use normalized point heights and intensity values to compute various descriptive statistics, generally called metrics, such as the mean, standard deviation, percentile heights, skewness, and kurtosis. These metrics are then used to fit models to predict field-measured attributes or used as attributes in classification approaches to identify species [6,9]. Early work presented by Brandtberg et al. in [11] used lidar metrics to distinguish leaf-off deciduous species. They report 38 to 50% correct classification accuracy using individual metrics computed using height and intensity, and 60% using multiple metrics. Further refinements and derivation of additional metrics from the point data presented by Brandtberg in [12] improve the accuracy to 64%. Orka et al. [13] and Liang et al. [14] classified deciduous and coniferous species using height and intensity metrics derived from discrete return lidar data. Orka et al. [13] achieved 88% accuracy for stem-mapped large trees using a combination of height and intensity metrics, and Liang et al. [14] reported 89% accuracy using a single metric: the difference between the first and last return surfaces for trees identified using a segmentation algorithm applied to a canopy height model. Li et al. [15] used high-density lidar data to classify four tree species (two coniferous and two deciduous) based on metrics describing structural features and reported an overall accuracy of 76%. More recently, Sun et al. [16] tested various deep learning methods and random forest (RF) to classify three species using high-density UAS (uncrewed aircraft systems) lidar and achieved up to 88% accuracy. Qian et al. [17] characterized general crown shape to classify five species and achieved 91 to 96% accuracy. While there is a significant amount of work related to species classification, there has not yet emerged a set of methods that can be universally applied across a broad range of forest types and species.

The shape of a plant is determined by the shape of the space that it fills. Most plants, including trees, attain a characteristic shape when grown in the open, free of competition [18]. Moreover, when grown under similar conditions, the general shape of the crown is relatively constant for a given species and individuals of the same age [19]. Brandtberg et al. [11] argue that the vertical distribution, configuration and features of branches and leaves (leaf-off conditions in their study), captured by common statistical metrics computed using the point clouds for individual trees, could be species dependent and, thus, could be used to train a classification method. However, branch and foliage development can be extremely plastic, resulting in variations in branch and foliage arrangement both within species and within individuals [19]. This plasticity presents a challenge when using crown morphology to identify species. Even if branch and foliage components, represented by points in lidar data, can be isolated and linked to form individual trees, as demonstrated by Hackenberg et al. [20] and Harikumar et al. [21], there may be so much variation in the arrangement of these components for an individual species that species identification remains problematic. In addition, physical damage to trees due to factors such as wind, ice or insect damage can also distort crown shapes.

The majority of the lidar data collected in the United States for forestry applications have been airborne scanning lidar from fixed-wing aircraft [22]. As of January 2022, such data are available or in progress for 83% of the continental United States [22]. Compared to traditional airborne lidar systems, lidar sensors carried by UAS cost less, have lower acquisition costs, produce higher density point data, and can be quickly and repeatedly deployed to collect data for small areas (10 to 100 s of hectare). While nationwide data are available for little or no cost, these characteristics make UAS lidar particularly well suited for operational planning and the monitoring of forest management activities. The point densities typical of UAS lidar data offer the potential for detecting differences in overall crown shape, branch geometry, and foliage distribution, useful for differentiating species. However, methods that strive to detect detailed branch and foliage patterns for individual trees, such as that described by Cárdenas et al. [23], are computationally expensive and difficult to apply over large land areas. Point cloud metrics derived from high-density UAS lidar data, which are easily and quickly computed, offer the potential to capture differences in crown structure and may thus be useful when identifying species.

Our overarching objective is to develop a robust classifier that can distinguish between Douglas fir (*Pseudotsuga menziesii* Mirb. Franco) and western hemlock (*Tsuga heterophylla* (Raf.) Sarg.) in forest types common to the Olympic Peninsula in western Washington. Mixed stands containing these two species are common to this area. The two species have different stumpage values [24] as well as ecological importance; so, the ability to identify them is important for forest management. Our hope is that the classifier can discern the "droopy leader" common to western hemlock compared to the more erect leader common to Douglas fir (Figure 1) based on simple, easily computed, lidar point cloud height metrics. To accomplish this, we intend to:

- Accurately combine field and lidar data to produce training data with high confidence that field trees have been matched to lidar point data;
- Train random forest (RF) classification models to distinguish between two conifer species common to forests of the Pacific Northwest;
- Compare the performance of classification models trained using point cloud metrics computed using height, intensity, and both;
- Compare the performance of a RF model trained using a small subset of height and intensity variables to the performance of a model trained using all variables.

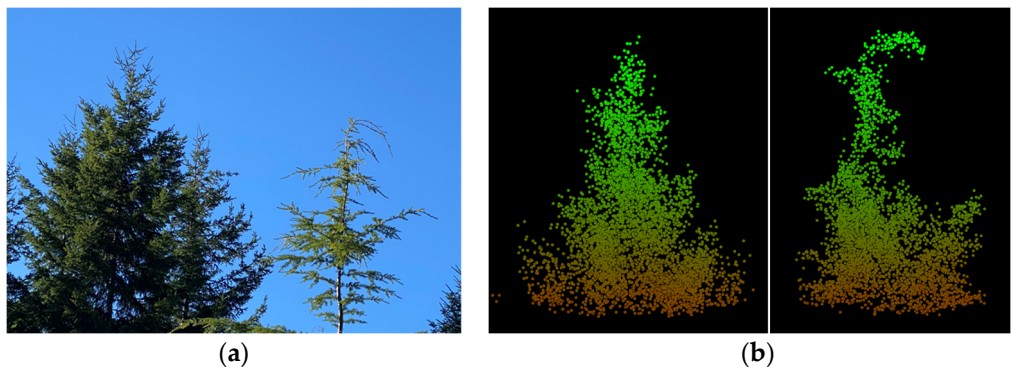

(**a**)             (**b**)

**Figure 1.** Photograph (**a**) and point clouds (**b**) showing the upper portions of crowns for Douglas fir (**left**) and western hemlock (**right**).

We found that the UAS lidar point data and derived metrics captured morphological and radiometric (represented by intensity metrics) differences in the upper canopy for our two species. These differences allowed us to produce RF classification models to classify species with accuracies of 91.8%, 88.7%, and 78.6% for models that included both height and intensity metrics, only height metrics, and only intensity metrics, respectively. A model trained with a subset of five metrics performed almost as well as the model trained with all metrics, with an accuracy of 91.5%.

## 2. Data and Methods

### 2.1. Study Site

The study site is located on the Olympic Experimental State Forest (OESF) on the Olympic Peninsula in western Washington (Figure 2). The OESF is managed through an experimental, integrated management approach by the Washington Department of Natural Resources (WADNR) to produce revenue for trust beneficiaries such as counties and public schools (primarily through timber harvest), habitat for threatened and endangered species and healthy streams for salmon and other aquatic species [25]. The specific harvest units involved in this study are part of the 8100 ha Type 3 Watershed Experiment (T3) designed to examine new, innovative approaches to managing forests to provide three kinds of possible benefits to trust beneficiaries [26]:

- High net revenue (but not necessarily maximum) within habitat conservation plan sidebars;
- Science-based learning focused on trust management issues;
- Increased public and tribal support for the management of trust lands.

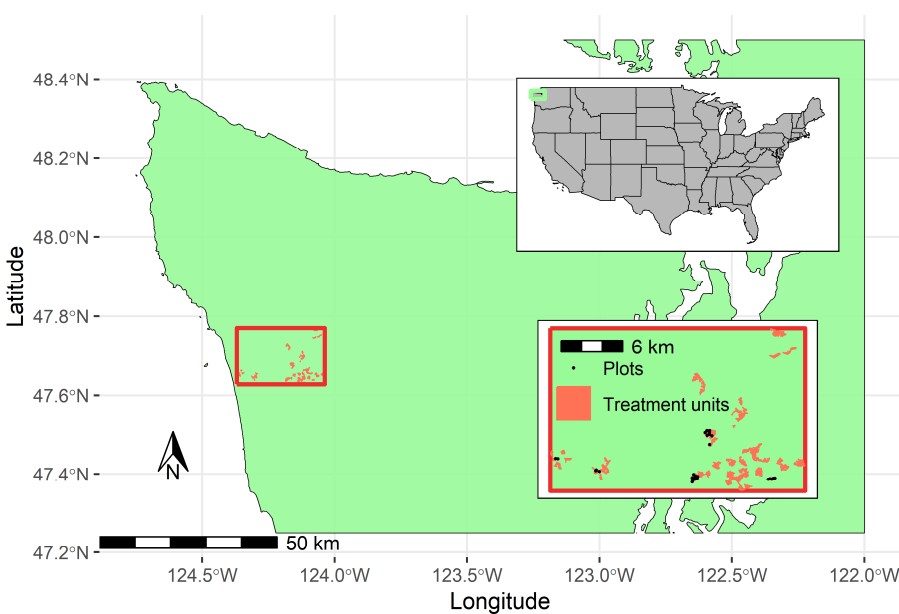

**Figure 2.** Study site located on the Olympic Peninsula in western Washington, USA. Field plots are located within treatment units that are part of the T3 Watershed Experiment.

The elevation in the T3 watersheds ranges from 41 m to 1034 m. The area is characterized by a maritime climate and receives heavy precipitation ranging from 203 to 355 cm per year, with the majority falling as rain during the winter. The watersheds are within the western hemlock climax vegetation zone, which is dominated by western hemlock. Variable amounts of Sitka spruce (*Picea sitchensis* Bong. Carriere), western redcedar (*Thuja plicata* Don ex D. Don), and Douglas fir. Silver fir (*Abies amabilis* (Dougl. ex Loud.) Dougl.) can be common at higher elevations and red alder (*Alnus rubra* Bong.) is common at lower elevations and near waterways. Vine maple (*Acer circinatum* Pursh.) and cascara buckthorn (*Frangula purshiana* (DC.) A. Gray) are present in understory and mid-story canopy positions. The area is characterized by young glacial and sedimentary soils, abundant moisture, and a long growing season, which result in rapid tree growth. The treatment units involved in this study are between 40 and 60 years old.

### 2.2. Field Plots

Tree measurements were collected on 27 circular plots covered by lidar data during the summer of 2021. The plot radius was 17.68 m. For each tree with a diameter at breast height (DBH) larger than 3.5 cm (measured on the uphill side of the tree 1.37 m above the ground), the following measurements were collected: tag number, species, DBH, distance from plot reference point, and azimuth from the plot reference point. Additional information was recorded to indicate whether the trees were dead, had excessive lean, a forked stem, or a broken top. One or more of these conditions disqualified a tree for use in training our classification model. A subjective evaluation was made and recorded for each tree to indicate if the tree was visible from above. This was carried out to identify trees that would likely be good candidates for pairing with tree objects identified using lidar data. Tree heights were not measured. Field data were recorded on paper and entered electronically by two people. Electronic copies were compared, and differences were resolved based on the paper copies or personal recollection by those who had performed the measurements. Measurements were collected on a total of 1528 trees (1428 Douglas fir or western hemlock). Figure 3 shows the number of trees by species and distribution of DBH for all trees. Of these, 741 trees were Douglas fir (338 trees) or western hemlock (403 trees), had no disqualifying conditions, and were considered visible from above.

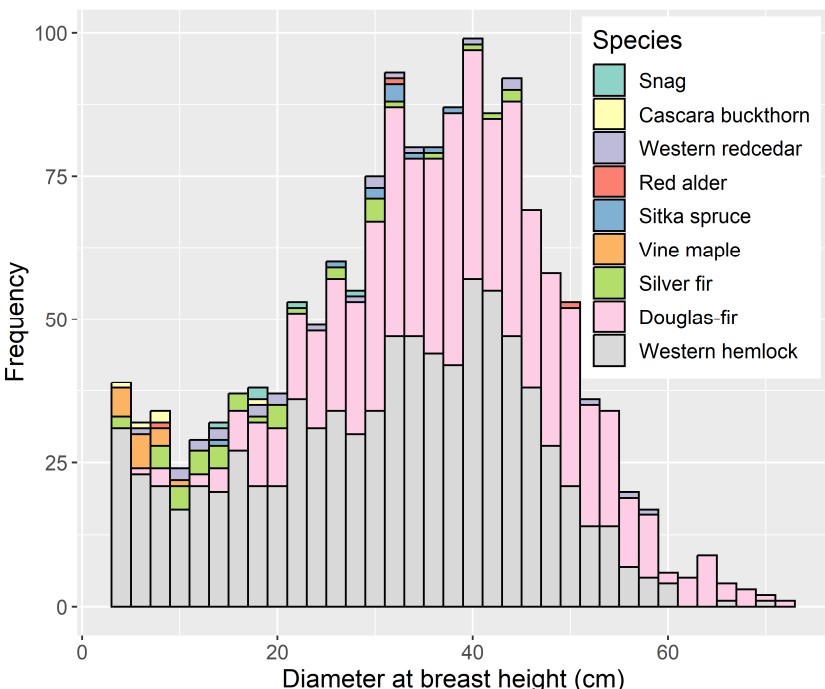

**Figure 3.** Species and diameter distribution of the 1528 trees measured on all plots.

The location of the plot reference point was obtained using Javad Triumph 2 GNSS receivers (San Jose, CA, USA). Reference points were trees that could be seen from all parts of the plot. For some plots, additional position data were collected at locations with less tree cover and a distance and azimuth recorded to the plot reference point. Position data were collected for at least 30 min, with positions recorded every second. Position data were post-processed for differential correction using a nearby continuously operating reference station. Previous experience in similar forest conditions with a similar GNSS protocol indicates that the positions collected are generally within two meters of the correct position [27,28]. However, larger errors are not uncommon given the dense canopy, large bole sizes typical of mature conifer forests in the Pacific Northwest region of the United States, and rugged topography.

### 2.2.1. Adjusting Plot and Tree Locations

The post-processed position for the plot reference point and the distance and azimuth measurements for tree bases were used to create a stem map and tree objects for each plot. Preliminary tree heights for the field trees were estimated using equation 4.1.1 and location code 609 from the Pacific Coast variant of the Forest Vegetation Simulator [29]. For all trees measured on the 27 plots, the average predicted height was 23.4 m with a standard deviation of 8.1 m. For the 741 Douglas fir and western hemlock trees, the average predicted height was 27.2 m with a standard deviation of 4.7 m. Final tree heights were measured in the lidar point cloud. These stem maps were compared to the lidar point cloud and canopy height model (CHM). Features in FUSION and the Lidar Data Viewer (LDV) specifically designed to facilitate the adjustment of plot and individual tree positions [30] were used to make overall adjustments to plot locations by aligning the largest or most prominent trees in the field data with trees visible in the lidar point cloud or CHM. Then, the locations of individual trees were adjusted to account for errors in the field azimuth or distance and tree lean. The tree movement features in LDV allow adjustment of both the base and top of trees to account for tree lean and to directly measure tree height. Figure 4 shows an example of the adjustments for a plot and an individual tree. In many cases, it was possible to align the stem of the modeled tree with returns from the stems in the point cloud. However, some trees did not have returns from their stems or only had returns from

the upper portion of their stems. For these trees, the modeled tree was first aligned with returns from the upper crown of the tree. Then, a manual iterative process was used to adjust the location of the tree base. Adjustments for all plots (741 trees) were performed by two analysts and the results were compared to assess the repeatability of the adjustments and to select the final set of field trees for use in model development. Trees for which the difference in horizontal location and height between the two analysts was within one meter were used for model training.

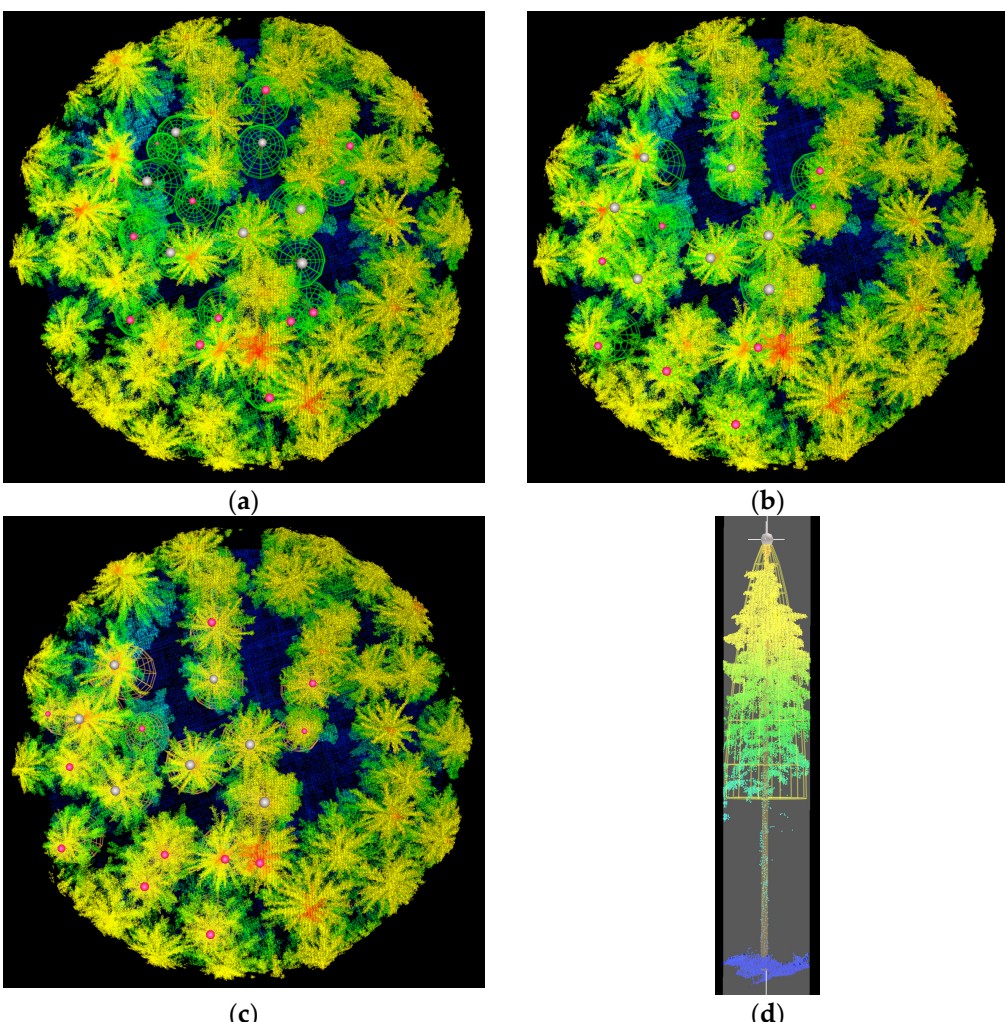

**Figure 4.** Screenshots showing plot 13 and tree location adjustments. (**a**) Shows an overhead view of point cloud and stem map produced using the post-processed GNSS position (without adjustment). (**b**) Shows an overhead view of point cloud and stem map after adjusting the plot location approximately 6.4 m to the southwest. (**c**) Shows an overhead view of point cloud after adjusting individual tree locations. (**d**) Shows a view of the points for a single tree and a modeled tree after adjusting the location of the tree base and treetop. For all panels, trees with grey markers (spheres) are trees that were Douglas fir or western hemlock, had no unusual conditions, and were considered visible from above. Trees with red markers (spheres) are other trees measured on the plot. Points are colored based on height, with cooler colors representing lower height points.

### 2.3. Lidar Data

Lidar data were collected using a Quanergy M8 (San Jose, CA, USA) sensor mounted on a DJI matrice 600 Pro hexacopter (Shenzhen, China) during the summer of 2021. The M8 sensor emits 430,000 905 nm pulses/second and collects up to 3 returns per pulse with intensity values for each return. The UAS was flown at a height of 50–80 m above

ground with the actual height dependent on vegetation height and topography. Data were collected for areas, hereafter called treatment units, ranging in size from 4 to 61 ha and containing one or more plots using two sets of parallel flight lines offset by 90 degrees for each area. This flight configuration resulted in a ground-level pulse density for a single swath of 90 pulses/m$^2$ (standard deviation: 104 pulses/m$^2$) and an aggregate pulse density of 556 pulses/m$^2$ (standard deviation: 519 pulses/m$^2$). Point data were filtered by the contractor to identify ground returns. Deliverables for the data included the classified point cloud, ground and highest hit surfaces at 0.5 and 1 m resolutions, and 1 m contour lines.

A preliminary examination of the lidar data revealed that individual tree crowns were scanned on all sides and the ground point density, while highly variable, was sufficient to create a high-quality ground surface. When visualizing the point cloud data, intensity values did not appear to provide any information related to tree species. However, an examination of the differences in the distribution of intensity metrics for each species indicated that intensity metrics could be useful in a classification model. Figure 5 shows violin plots depicting the distribution of several height and intensity metrics, providing evidence that both types of metrics may have value for species classification. The median values shown in Figure 5 (horizontal lines) are all significantly different, as determined using Mood's median test as implemented in the coin package 1.4-3 [31] in R 4.3.2 [32].

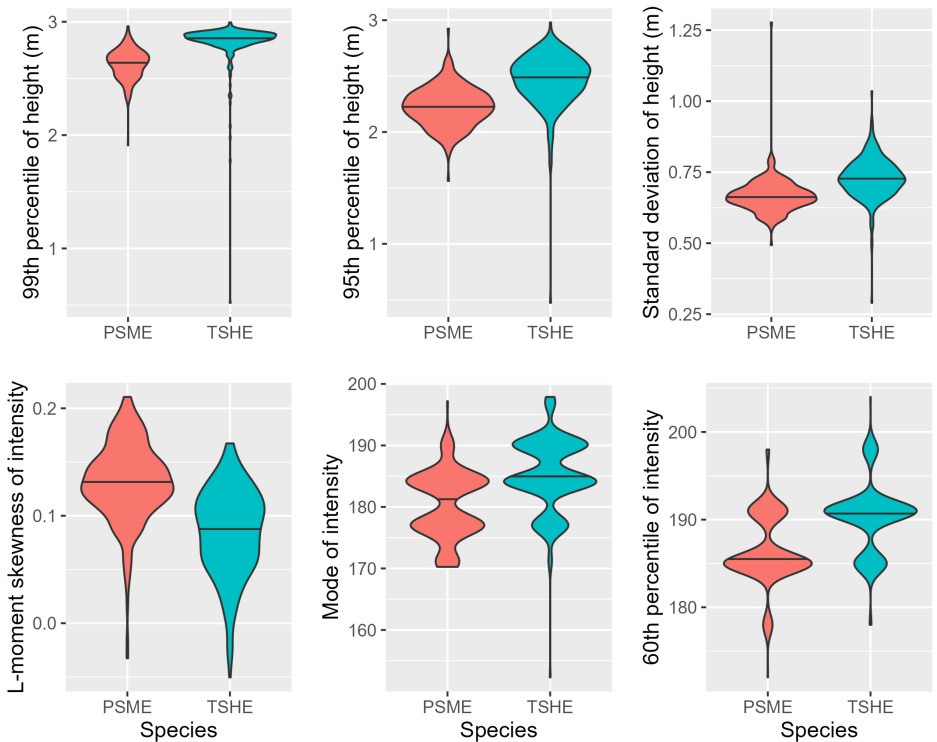

**Figure 5.** Violin plots showing the distributions of three lidar height metrics and three intensity metrics by species Douglas fir (PSME) and western hemlock (TSHE). Horizontal lines within the shaded regions indicate the median value for each species. Intensity values from the Quanergy M8 sensor are discretized leading to the irregular distributions seen in the plots depicting the mode and 60th percentile distributions; however, the details of the discretization logic are unknown.

Lidar Data Processing

Point cloud data were processed using a combination of FUSION 4.5 [30] and R 4.3.2 [32] with the lidR package 4.0.4 [33,34] and the fusionwrapr package 0.1.0 [35].

Using the adjusted tree base and top locations for the trees selected for model development, a cylinder of points (one meter radius) was clipped from the point cloud for each tree using the lidR package to clip points within a bounding box that encompassed the leaning tree. R code was then used to clip points in the leaning cylinder. The points in the

upper three meters of the clipped data were extracted and normalized using the elevation of the highest point in the clip minus three meters. This process of normalization using the same reference height for all points prevents distortion to crown morphology that occurs when normalizing using an interpolated ground elevation for each point (Figure 6). The normalized points were used to compute a suite of metrics using FUSION's CloudMetrics tool. In addition, relative height percentile values were also computed in R (percentile heights divided by 99th percentile height). Table 1 describes the metrics.

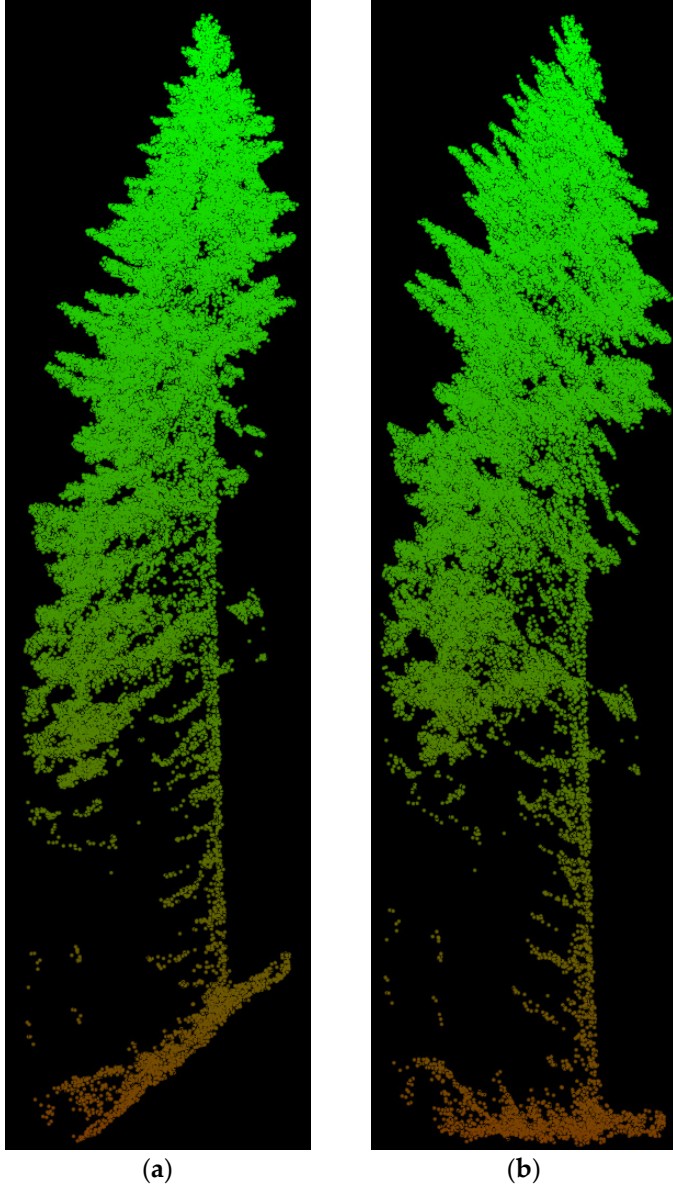

(**a**)                    (**b**)

**Figure 6.** Point cloud for a single tree located on a ~100% (45 degree) slope without normalization (**a**) and with normalization (**b**). Normalization was done by subtracting a ground elevation interpolated for each point XY location from the point elevation. Normalization using a single elevation value for all points maintains the same crown and branch morphology as the tree points without normalization. The single elevation value could be the elevation at the tree base or, in our case, the height minus three meters.

**Table 1.** Point cloud metrics computed for points in the upper three meters of a one-meter radius cylinder centered on the treetop location using FUSION's CloudMetrics tool. The "X" in the **Computed for Height** and **Computed for Intensity** columns indicates that the metrics was computed for these point attributes.

| Metric Name | Computed for Height | Computed for Intensity | Description |
|---|:---:|:---:|---|
| Minimum, maximum, mean, mode, standard deviation, variance, coefficient of variation, interquartile distance, skewness, kurtosis, average absolute difference | X | X | Standard descriptive statistics. Minimum and maximum height were dropped from the set of variables because we expect these to have nearly the same values for all trees: minimum = 0 and maximum = 3. |
| MAD.median | X | | Average distance between each data point and the median. |
| MAD.mode | X | | Average distance between each data point and the mean. |
| L-moments | X | X | L moments are computed using linear combinations of ordered data values (elevation and intensity) [36]. The first four L moments (L1, L2, L3, L4) are estimated using the direct sample estimators proposed by Wang [37]. L1 is exactly equal to the mean. |
| L-moment ratios | X | X | Ratios of L moments provide statistics that are comparable to the coefficient of variation (L2/L1), skewness (L3/L2) and kurtosis (L4/L2). |
| Percentiles | X | X | Height or intensity value below which a given percentage, k, of values in the frequency distribution falls. k = (1, 5, 10, 20, 25, 30, 40, 50, 60, 70, 75, 80, 90, 95, 99). |
| Canopy relief ratio | X | | $\frac{(\text{mean height}-\text{minimum height})}{(\text{maximum height}-\text{minimum height})}$ |
| SQRT.mean.SQ | X | | $\sqrt{\frac{\sum \text{height}^2}{n}}$ |
| CURT.mean.CUBE | X | | $\sqrt[3]{\frac{\sum \text{height}^3}{n}}$ |
| Profile area | X | | Modified version of the area under the height percentile curve described by Hu et al. [38]. Modifications to the calculation method are described in McGaughey's work [30]. |
| Relative percentile heights | X | | Percentile height, k, divided by the 99th percentile height with k = (1, 5, 10, 20, 25, 30, 40, 50, 60, 70, 75, 80, 90, 95). |

To facilitate the application of our species classification model, a CHM was produced with 0.5 m resolution. The CHM was smoothed using an average filter applied over a three-by-three cell window. CHMs were created using all data for each treatment unit, leading to an area much larger than the individual plots. Using FUSION's TreeSeg tool, individual tree objects (ITO) were segmented from the CHM with a minimum height threshold of 2 m (areas of the CHM below 2 m were not considered as candidate areas for segmentation). The ITO highpoint locations produced by TreeSeg were used to clip a cylinder (one-meter radius) of points for each ITO. The full suite of metrics shown in Table 1 were computed using points in the upper three meters of each cylindrical clip.

### 2.4. Model Development

All of the final training trees were used to tune the hyperparameters (mtry, min.node.size, and sample.fraction) of RF models with 1000 trees using the tuneRanger package 0.5 [39] in R with prediction accuracy as the measurement used for optimization. While Probst and Boulesteix [40] argue that the performance gains that result from using 1000 trees with RF are small compared to those using 250 trees, the computational time to fit and validate our model was not excessive; so, we opted to use more trees. Separate models were tuned and fit using

all point cloud metrics, only metrics involving intensity (no height metrics), and only metrics involving height (no intensity metrics). After tuning, the final models were fit using the tuned hyperparameter values and all tree data using the ranger package 0.16.0 [41]. Leave-one-out cross-validation (LOOCV) was then performed using the tuned hyperparameter values to evaluate the prediction accuracy associated with each set of metrics.

We also tested a model trained using a subset of variables and evaluated its prediction performance using LOOCV. The subset was selected using variable permutation importance scores from the model that used all variables. Importance scores were sorted from highest to lowest. We computed a matrix of the absolute value of Spearman's rank correlation coefficient (hereafter called correlation) between all variables, sorted in the same order as the importance scores, and extracted the lower triangle, excluding the diagonal. To assist with the selection process, the maximum correlation for each row of the matrix was computed. This maximum value represented the highest correlation with variables that had higher importance scores than the variable represented by the row in the matrix. Our selection process, designed to exclude the least important of a pair of moderate to highly correlated variables, as defined in [42], started by selecting the variable with the highest importance score. Then, the maximum correlation value was checked for the variable with the next highest importance score. If the maximum correlation with other variables with higher importance scores was lower than a threshold value (0.5 in our analyses), the variable was included in the subset. This process was repeated for all variables to produce a subset of variables used to fit, tune, and evaluate another RF model.

### 2.5. Model Application

The metrics computed for the upper three meters of each one-meter radius cylinder centered on the ITO highpoints were used with the best model, based on the lowest prediction error, to predict species for each ITO, and the results were mapped for each of our treatment units.

## 3. Results

### 3.1. Confidence in Linking Field Stem Data with Lidar Point Clouds

Tree locations from field data were shifted using the two-step adjustment process described in Section 2.2.1. The repeatability of the process was confirmed by comparing the shifts made independently by two analysts (Figure 7a). For all but three plots, the overall adjustment for plot locations was less than two meters, with no major differences between analysts. Plots 9, 13, and 43 had horizontal errors in the post-processed GNSS positions based on the adjusted tree locations of 3.4 m, 7.3 m, and 2.1 m, respectively (Figure 7a). Such errors are most likely due to the steep topography and presence of large-diameter trees leading to non-line-of-sight and multipathing of GNSS signals [27,28]. For most individual trees, the adjustment difference for the tree base and top locations between analysts averaged well under 1 m (Figure 7b,c). The differences between top locations between analysts were generally smaller than the differences in the base locations. This was not surprising since treetops were more easily identified in the point cloud compared to tree bases. Outlier points in Figure 7b,c probably represent cases where field trees were matched to different trees in the point cloud by each analyst. Forest canopies on our plots were dense and it was sometimes difficult to see individual treetops in the point cloud and match field trees to the point data.

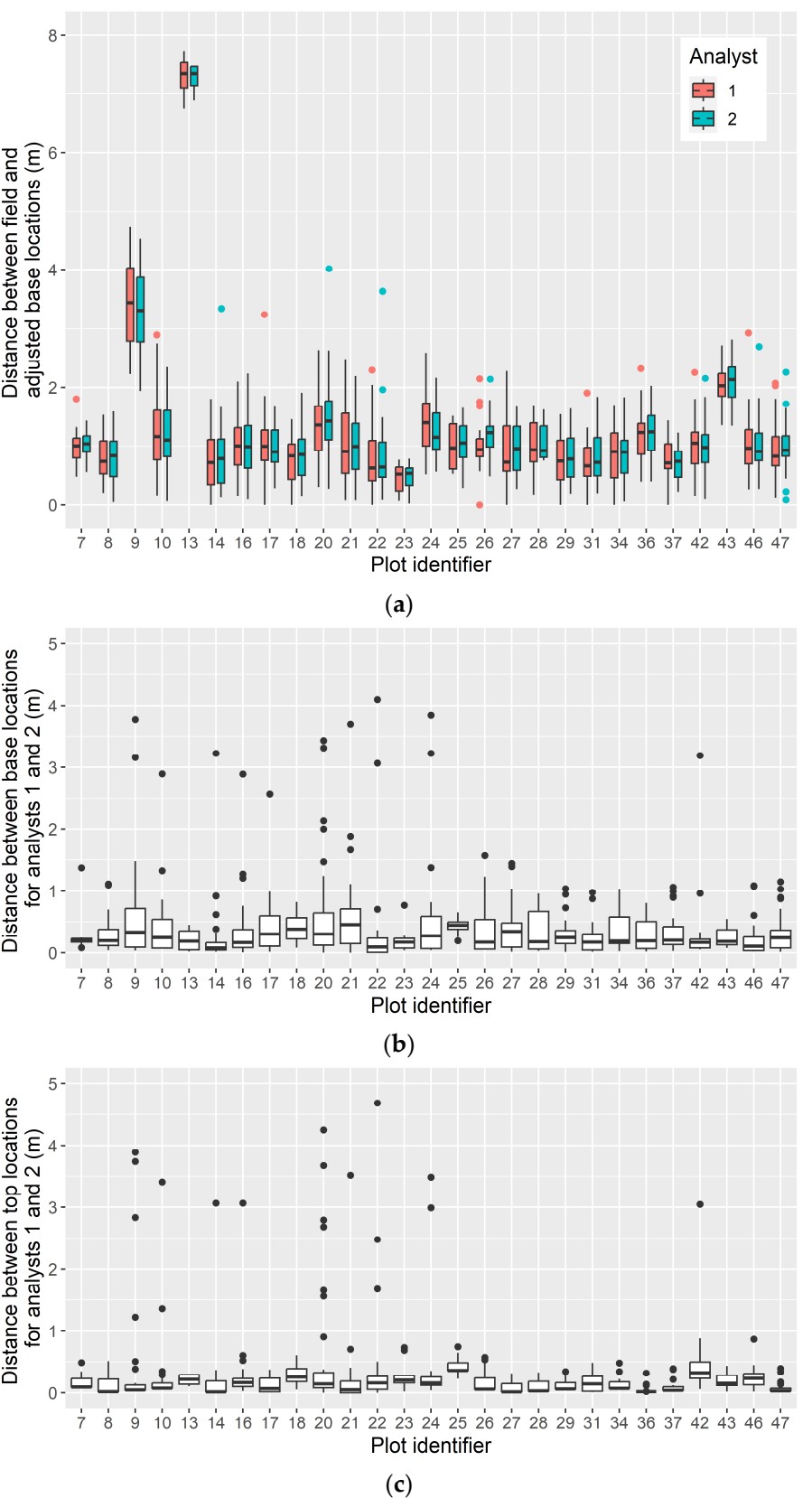

**Figure 7.** Comparison of the tree adjustments performed by two analysts. (**a**) Shows the horizontal distance between the original tree locations based on the post-processed GNSS positions for the plots and the adjusted base locations for the two analysts. (**b,c**) Show the distance between the adjusted locations for the tree bases and treetops for the two analysts, respectively.

Trees for which the horizontal location and height for the two analysts were within one meter were included in our model training data set (575 trees: 269 Douglas fir and 306 western hemlock). The base and top locations for the two analysts were averaged to produce the final position and orientation for each tree. Figure 8 shows the distribution of the species and tree diameters for the training trees.

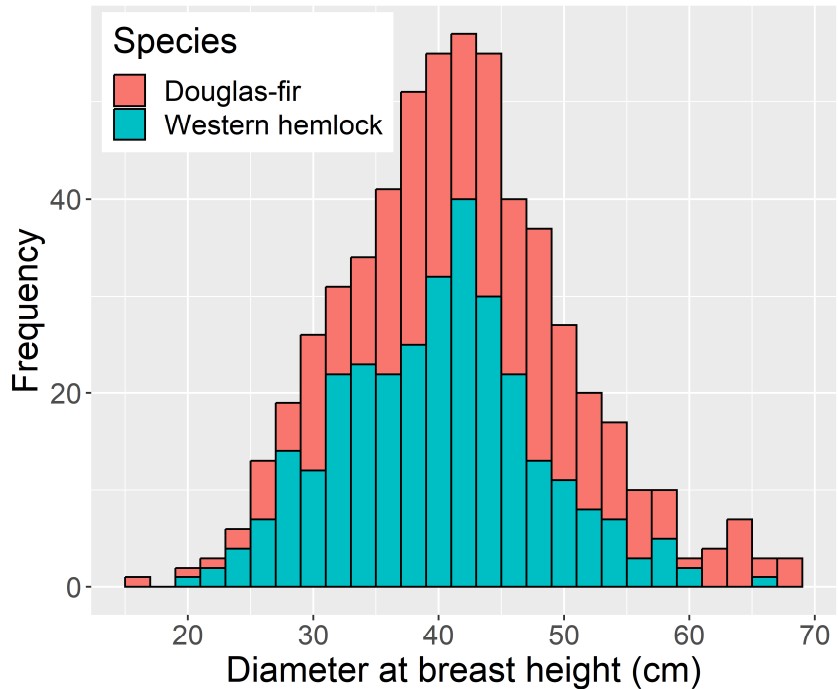

**Figure 8.** Species and diameter distribution of the 575 trees used for model training.

### 3.2. Model Tuning and Accuracy

Table 2 summarizes the values for the hyperparameters, resulting accuracy, and Cohen's kappa using all variables, only height variables, only intensity variables, and a subset of five variables. All accuracy values shown in Table 2 were produced using LOOCV. The subset of variables included, in order of permutation importance, 99th percentile height, L-moment skewness of intensity values ($L3_{intensity}/L4_{intensity}$), 60th percentile of intensity values, and the fourth ($L4_{height}$) and third ($L3_{height}$) L-moments of height. This subset includes the most useful variables, as measured by permutation variable importance, that are not highly correlated with other useful variables.

**Table 2.** Values for tuned hyperparameters, the resulting classification accuracy evaluated using leave-one-out cross validation, and Cohen's kappa.

| Predictors | mtry | min.node.size | sample.fraction | Accuracy (%) | Kappa |
|---|---|---|---|---|---|
| Height and intensity | 12 | 2 | 0.46858 | 91.8 | 0.83 |
| Height only | 22 | 18 | 0.20776 | 88.7 | 0.77 |
| Intensity only | 28 | 21 | 0.51976 | 78.6 | 0.57 |
| Subset | 1 | 3 | 0.20687 | 91.5 | 0.83 |

### 3.3. Model Application

Figure 9 shows the predicted species for a single treatment unit. While we did qualitatively compare the predictions with aerial photographs, we did not conduct any type of quantitative validation using aerial photographs. Douglas fir and western hemlock were not easily identified using the imagery available to us due to low resolution and deep shadows. We did conduct a visual evaluation of the ITO point clouds looking for crown morphology characteristics of our two species and found that, in general, species

predictions were consistent with the species determined from the point cloud. However, we did find several ITOs where species determination was not possible or was ambiguous due to visible crown damage (e.g., multiple leaders, leader damage), multiple trees included in the ITO (under-segmentation), and partial trees included in the ITO (over-segmentation). We also observed point clouds that showed morphological characteristics typical of other species present in our study area, such as Sitka spruce and western redcedar. Given that our model was trained using only data for Douglas fir and western hemlock, species predictions for trees of other species are obviously incorrect.

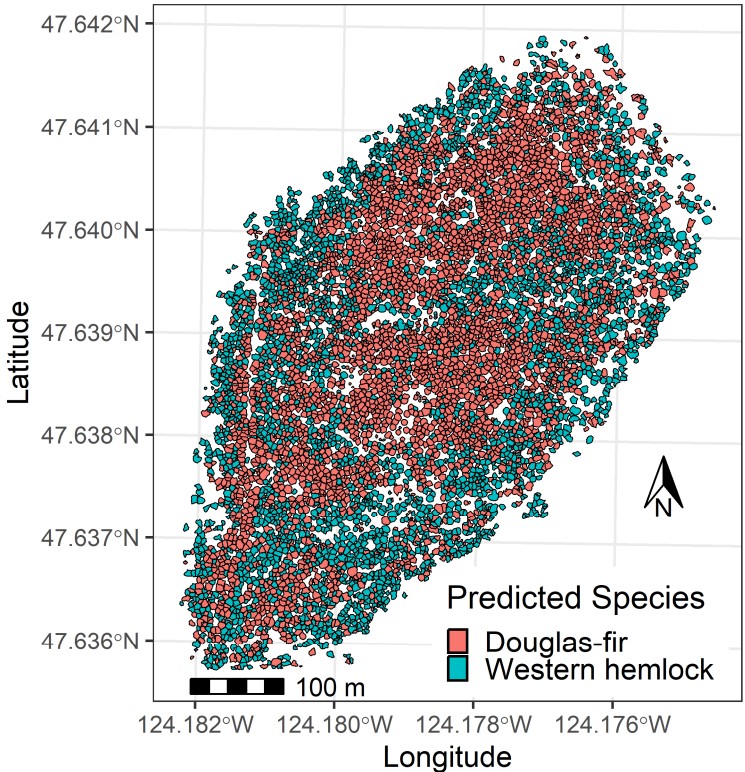

**Figure 9.** Map showing the classification model applied to data from ITOs to map species over a 27.8 ha treatment unit.

## 4. Discussion

In this study, we explored the potential of UAS lidar data to classify two species common in the Pacific northwest region of the United States. To ensure accurate matching between field trees and lidar data, we manually adjusted individual plot and tree locations aligning tree bases and tops with features that could be identified in the lidar point cloud. Our results showed that point cloud metrics for the upper three meters of a one-meter radius cylinder centered on the tree top captured information needed to train a random forest classification model to distinguish between Douglas fir and western hemlock trees with 91.8% overall accuracy using metrics derived from point heights and intensity. The accuracy was reduced using only height or only intensity metrics to 88.7% and 78.6%, respectively. Finally, a model fit using a subset of five height and intensity metrics performed almost as well as the model using all metrics (91.5% overall accuracy).

Our results regarding the utility of height and intensity metrics for identifying species are similar to those reported by other authors using similar methods. Lin and Hyyppä [43] evaluated metrics describing the point height distribution (PD), intensity (IN), and crown-internal (CI) and tree-external (TE) structural signatures to classify four species. Classification accuracies for the different variable types were PD (65%), IN (80%), CI (83%), and TE (85%). The best accuracy was achieved by selecting a subset of metrics from each group

(93%). Similarly, these studies report that their highest accuracies were obtained using combinations of height, intensity, and other metrics compared to using only one type of metric [11–13].

For our data, the intensity values from the Quanergy M8 sensor were stored as eight-bit integer values. However, the distribution of values was not continuous. There appeared to be some form of scaling applied to the intensity values, but details were not evident or available. Figure 10 shows a plot of intensity values for all returns from a single flightline. We suspect that the intensity scaling contributes to the reduced accuracy and lower kappa coefficient of the classifier fit using only intensity metrics (accuracy: 78.6%; and kappa: 0.57). While we did find that the intensity metrics improved the performance of our classification model, we wonder if these metrics would have been even more useful without the scaling.

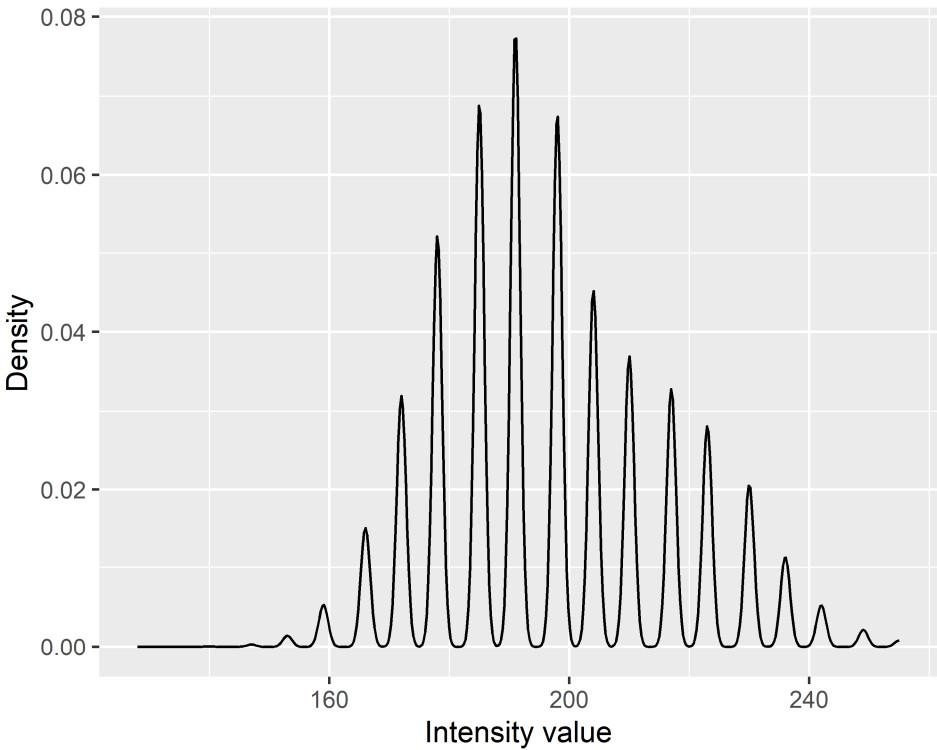

**Figure 10.** Distribution of intensity values for data from a single flightline showing the non-continuous distribution.

We computed intensity metrics using all returns. It might have been better to compute the metrics using only first returns. However, the upper 3 m of the crowns were composed of an average of 94.3% first returns (standard deviation: 3.4%); so, our computed metrics were most influenced by first returns. Future work will investigate the utility of intensity metrics computed using only first returns.

Unlike other studies, we limited the lidar points used to compute metrics to a small portion of the upper crown (the upper three meters of a one-meter radius cylinder centered on the tree top and adjusted to account for tree lean). By doing this, we feel we were successful in capturing the differences in shape and vertical foliage density for upper crowns of our two species of interest. The most important metric based on variable permutation importance scores from RF was the 99th percentile height. This makes sense given that western hemlock trees are characterized by a "droopy leader"; that is, the upper half of the leader is bent 60 to 90 degrees from vertical (Figure 1). This results in western hemlock having higher proportion of foliage, and thus more lidar returns, near the treetop compared to Douglas fir, making the 99th percentile height higher for western hemlock.

We limited our metrics to those computed using the distribution of point heights and intensity values. These metrics are easy to compute and not computationally demanding.



There could be additional metrics that might better capture the shape of the tree leader and individual branches. Hackenberg et al. [20] and Harikumar et al. [21] demonstrate approaches that identify individual branches from which additional predictive attributes could be derived. However, branch and foliage development can be extremely plastic, resulting in variations in branch and foliage arrangement both within species and within individuals of the same species [19]. A simpler and more computationally realistic approach for large areas would be to divide the points for the upper crown into radial segments and then compute height distribution metrics for each segment. These metrics and summary metrics derived from them might better capture the branching structure of the upper crown and, thus, facilitate higher classification accuracies. The pursuit of new point cloud metrics is a worthwhile endeavor but must be balanced with the computational demands presented by more complex methods. Developing metrics for a few hundred trees to provide training data for classification models is likely feasible, but producing those same metrics for all trees over large areas so the resulting models can be applied may be computationally onerous.

Our sample size was small and slightly unbalanced (269 Douglas fir and 306 western hemlock). The small sample size led us to select LOOCV over k-fold cross validation. We were concerned that the small sample size could lead to instability in the performance estimates obtained from k-fold cross validation. For our data set, LOOCV could be performed in a few minutes. For larger data sets, the computation time needed for LOOCV could make k-fold cross validation a more suitable option. The sample unbalance was dealt with by using the sample.fraction parameter for RF in the ranger package [41].

We limited our modeling approach to RF but acknowledge that there are other modeling approaches that may produce similar or better results. Possible methods include classification tree methods (e.g., decision trees, RF), grouping and separability methods (e.g., Support Vector Machine (SVM), k-nearest neighbors (kNN)), and deep learning methods (Convolutional Neural Network (CNN)). Michalowska and Rapiński [10] reviewed papers focused on using lidar data for species classification and found that RF and SVM performed similarly when both were applied using the same data. While they reviewed studies using all methods, no studies compared all methods using the same data; so, it is hard to select a "best" method based on their review. Given that our future research plans include multi-species classification, we were slightly worried by the following statement in Fassnacht et al.'s work [44]: "the optimal design of multi-class SVM is demanding". In addition, they point out that "From the experience of the authors, the choice of the classifier itself is often of low importance if the data is adequately pre-processed to match the requirements of the classifier". Korpela et al. [45] (not reviewed in [10]) report that RF outperformed SVM and other classification methods in the classification of the 21 mire site vegetation types in their study and had a similar performance to SVM for dominant species classification. We are comfortable with our choice of RF for our modeling but concede that there is a need to explore different classification approaches for future work.

We relied on ITOs derived from a CHM to apply our model over a larger area. However, a CHM was not used when developing the set of trees used for model training. We evaluated various resolutions (0.1–1.0 m cells) for the CHM with and without smoothing using a trial-and-error approach. The selected resolution (0.5 m cells) and smoothing (three by three average filter) produced a set of trees that captured the dominant and co-dominant trees without excessive over-segmentation. In general, finer resolutions resulted in smaller crowns for the dominant and co-dominant trees, and more over-segmentation and coarser resolutions produced a set of trees very similar to those produced using the 0.5 m CHM. Without smoothing, the CHMs produced from the point cloud contained too much branch detail and resulted in over-segmentation. Smoothing applied to the CHM reduces the overall height of the CHM. However, we only used the horizontal location of the segmented tree high points to locate our sample cylinders so the absolute accuracy of tree heights was not essential. There are more complex methods for creating, evaluating, and selecting CHM resolution and smoothing. Mielcarek et al. [46] evaluated the impact of CHM interpolation methods on the accuracy of tree height estimates. Zhao and Popescu [47] compare the

effect of CHM smoothing on the detection of individual treetops. However, since our goal was to simply find highpoints associated with overstory trees that would be fully, or nearly fully, visible to a lidar scanner mounted on a UAS so we could test our classification model over a large area, we did not attempt to further refine or optimize our CHM.

Crown morphology changes and becomes more variable as trees get older. However, the crown shape stays consistent until trees reach 150–200 years old [48]. All stands in our study were about the same age (40–60 years) and had experienced similar levels of disturbance; so, we expected crown morphology to be consistent from tree to tree provided there had been no physical damage to the upper crown. While the lidar metrics used for our classification model seem to capture the shape of the upper crown well, the utility of our approach might be reduced in areas with older trees or more crown damage.

Our study is limited to two species: Douglas fir and western hemlock. There were more species present within our study area (Figure 3) but our field data from 2021 contained too few trees of other species to train a model. Of the 1528 trees measured on all plots, only 100 were species other than Douglas fir or western hemlock. Of these 100, only 33 were labeled by field crews as visible from above. When selecting training trees, we matched field trees to trees visible in the lidar point cloud. The plot/tree adjustment process, described in Section 2.2.1, gave us high confidence that all trees used to train the classification model were either Douglas fir or western hemlock. When applying our classifier over a larger area, the trees identified by our segmentation tool using a CHM most likely correspond to trees that would be marked as visible from above. We acknowledge that some of the segmented trees may be species other than Douglas fir or western hemlock. However, the proportion of other species in the upper canopy, based on our plot data, is small. Nonetheless, the only species that can be assigned by the classifier are Douglas fir and western hemlock; so, any other species would be mis-classified. To help address this deficiency, we collected more field and lidar data (using a different lidar sensor) in 2022 and 2023 along with multispectral imagery over some plots in 2023. We plan to expand our analyses to include more species and to explore the utility of data from the two lidar sensors and the multispectral imagery for species classification.

Our UAS lidar data were collected late in the growing season (September). The shape of the upper crown, particularly the leader, and branch attributes may be different at different times of the year. The bent leader, typical of western hemlock, is visible throughout the year but the leader may be more erect during the period of peak growth. Additional data, collected early in the growing season, would be needed to learn whether differences in morphology affect the classification accuracy.

Our UAS lidar data were of high density (556 pulses/m$^2$). While this is typical of UAS lidar, the density is much higher than that of the data commonly collected in large-area campaigns [22]. Based on qualitative visual assessments, the point cloud data captured individual branches and the leaders of trees visible from above. Lower density data would likely provide less detail for these features and the classification accuracy would decrease. Compared to lower density data captured from higher altitudes, our UAS lidar is noisier. This is partially due to lower quality GPS and inertial measurement sensors, but most of the noise seems to come from tree movement. Most of the trees in our study area were imaged by data from several flightlines (6–8 passes over individual trees was not uncommon). Any tree movement due to wind causes slight shifts in the points from the different flightlines. While this movement does not have a major impact on the computed metrics, it does cause problems for approaches that try to identify individual branches. For such methods, it may be desirable to treat the data from each flightline separately and merge the results or attempt to align the data from multiple flightlines prior to any further processing.

Studies that work with individual trees must pay attention to the alignment of field-measured and remotely sensed data. Tree positions from field data are usually based on the location of the base of the tree. However, nearly all trees exhibit some amount of lean; so, the horizontal position of the treetop is different from the tree base. Most studies that match lidar data to individual trees carry this out by finding the closest lidar-derived tree

to a field-measured tree or vice versa. In our forest type, trees occur close to one another; so, this matching logic does not always produce correct matches. For classification purposes, if the tree species are the same, the matching error is not a problem. However, if the species are different, this produces errors in the training data and results in lower overall prediction accuracies. Our plot and tree adjustment process, performed by two separate analysts and compared, provided us with confidence that field trees were correctly matched to trees visible in the lidar point data. Alignment methods that rely solely on GNSS positions for plots or trees may not produce the same level of confidence.

Species classification efforts using lidar data benefit from additional remotely sensed data such as aerial imagery. While lidar intensity provides some indication of spectral reflectance, multi- and hyperspectral imagery provide a more useful spectral signature to differentiate tree species. Hell et al. [49] demonstrate the use of lidar and multispectral imagery to classify broad species groups and dead trees using deep learning methods. Zhong et al. [50] combined lidar data and hyperspectral imagery to classify five tree species. Both groups found significant improvements in the classification performance when metrics from imagery were used in conjunction with lidar metrics. It is clear from the literature that imagery, either multispectral or hyperspectral, provides valuable information for species classification. However, the addition of imagery adds to the cost of data acquisition and limits the time of day when data can be collected. In addition, extra care must be taken to process imagery so that positional errors and image distortion are minimized.

Using UAS lidar data to distinguish tree species is useful for planning and monitoring forest management activities. UAS systems offer advantages compared to traditional airborne lidar in that they can be deployed quickly to cost-effectively collect data for relatively small areas. While high-density UAS lidar has been used by other researchers to characterize detailed crown structure, our approach, which uses easily computed point cloud metrics, is not computationally intensive, making it suitable for application to forestry projects ranging in size from 10 s to 100 s of hectares. The information products derived from UAS lidar can augment or, in some cases, replace field data commonly used to plan and monitor forestry activities. However, our study only addresses a small part of the information useful to foresters. Future studies could expand on our methods to identify additional species and other useful forest metrics.

## 5. Conclusions

Our study demonstrated the ability of UAS lidar height and intensity metrics to distinguish between 40- and 60-year-old Douglas fir and western hemlock trees when used to fit a random forest classification model. This ability is important given that these two species frequently occur together in the region represented by our study and have different ecologic and economic values. Our procedure for aligning field-measured trees with the lidar point cloud produced reliable data for training our model. The procedure demonstrated a relatively easy approach to correct errors in plot location common in the forests common to the Pacific Northwest, where dense canopies and large stems can cause errors in GNSS positions. As an additional benefit, this procedure produces information describing the lean for individual trees, which could be useful in other studies. The small sample of lidar point data representing the upper 3 m of each tree crown and the resulting height and intensity metrics was sufficient to capture morphological and radiometric differences between the two species. Using a combination of 86 lidar height and intensity metrics, we were able to fit a RF model that achieved 91.8% overall accuracy. We also fit a model with a subset of five metrics that performed nearly as well (91.5% overall accuracy). Finally, we applied our model to classify species over entire study units, demonstrating that the model could produce results that should be useful for silviculture planning and economic analyses.

**Author Contributions:** Conceptualization, R.J.M., A.K., C.R.B. and B.T.B.; methodology, R.J.M.; software, R.J.M.; validation, R.J.M., A.K., C.R.B. and B.T.B.; formal analysis, R.J.M.; investigation, R.J.M., A.K., C.R.B. and B.T.B.; resources, R.J.M.; data curation, A.K. and C.R.B.; writing—original draft preparation, R.J.M.; writing—review and editing, R.J.M., A.K., C.R.B. and B.T.B.; visualization, R.J.M.; supervision, C.R.B. and B.T.B.; project administration, R.J.M. and B.T.B.; funding acquisition, R.J.M. and B.T.B. All authors have read and agreed to the published version of the manuscript.

**Funding:** This research was funded, in part, by the University of Washington's Olympic Natural Resources Center, with support from the Washington State Legislature and the Washington Department of Natural Resources. This research was conducted at the Olympic Experimental State Forest managed by Washington State Department of Natural Resources. In addition, this research was supported by the U.S. Department of Agriculture, Forest Service.

**Data Availability Statement:** The model training data, lidar point data, and ground models for plots are available from https://github.com/bmcgaughey1/UAS_Lidar_Species_Classification.

**Acknowledgments:** The authors would like to thank the interns at the University of Washington's Olympic Natural Resources Center for collecting the field data used in this study. We would also like to thank Chris Erickson of West Fork Environmental Inc. for partnering with us to collect and prepare the lidar data used in this study.

**Conflicts of Interest:** The authors declare no conflicts of interest.

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
