# Peer review of "Tree Species Classification Based on Upper Crown Morphology Captured by Uncrewed Aircraft System Lidar Data"

_remotesensing, doi:10.3390/rs16040603_

Round 1

Reviewer 1 Report

Comments and Suggestions for Authors

Dear authors,

Thank you for this study. I read the manuscript with great interest. Please find below some minor comments.

L18 I think '(RF)' not needed

L48 consider to write author’s name instead of 11

L53 I’d welcome a relatively early explanation of ‘intensity metrics’. Probably briefly explain both, height and intensity metrics.

L144 and following. I would welcome an assessment and/or a discussion on target vs. other tree species. How does the occurrence or abundance of other tree species than Douglas-fir or western hemlock (sitka spruce, western red cedar, silver fir, red alder) influences model results?

Consider a blank at 'redcedar'

Since many analyses address height metrics, I’d be curious to know how high the study trees were.

L166 ‘unusual conditions’ please explain what that means.

I’d also welcome a discussion on how this reduction from 1528 overall measured trees to 624 trees considered (no unusual conditions and visible from above) may influence model performance and in particular application of the study results.

L216 above ground (?)

Reviewer 2 Report

Comments and Suggestions for Authors

The manuscript titled "Tree species classification based on upper crown morphology captured by UAS lidar data"is a well written, needed piece of research in the UAS LiDAR space. The authors make that clear in the robust introduction. There is ample space for better classification, and UAS LiDAR can be the way forward. A RF classifier was used to differentiate between species, and the models very well. 

While the language and written document is fine. I have a few questions about the work and why certain approaches were taken. 

Why use the random forest model rather than a support vector machine or other machine or deep learning classification model?

Why no conclusion section? I believe it would enhance the paper by providing a final statement on the importance of the findings.

Why was the CHM at 0.5 meter resolution, as well as the products from the contractors? With such high density that UAS LiDAR data can provide, I believe a. much higher CHM can be created, which may improve accuracy. That is one benefit of UAS LiDAR. 

Thank you for a great manuscript and progress in the field.

Reviewer 3 Report

Comments and Suggestions for Authors

Review comments:

1. In Figure 1 (b), only radar data from one angle is displayed. Douglas seems to have similar data for each angle. What does Western Hemilock look like for radar data from other angles? How can this study better capture the bent branches? Please add a schematic diagram and explain.

On line 2.210, the red dots represent other tree species. The classification effect of Figure 9 does not include other tree species. Has the other tree species been omitted or are they included in the two tree species in this article?

3. This study selected point cloud data within a range of 3m from the tree canopy as the input. Whether to consider other ranges, such as 5m, and whether the classification accuracy has changed, further experiments were conducted to demonstrate the feasibility of selecting 3m.

4. The classification model using height and intensity indicators achieved an overall accuracy of 91.8%. How can future research introduce more features to distinguish more tree species and improve the model's generalization ability?

What is the classification accuracy of low resolution aerial photos from lines 5.347 to 349? Can obtaining high-resolution aerial photos in the future make it easier to distinguish tree species and compare classification results with using radar data.

Line 6.111, missing content after and

7. Discussion section: Further explore the impact of using LiDAR data with different data quality and resolution on research results; The impact of ground truth on LiDAR data collection and tree species classification. In addition, it is possible to explore more tree species classification, applications in different regions, and data collection in different seasons. Consider possible technological innovations, such as combining other sensor data, improving machine learning models, etc., to improve the accuracy and applicability of tree species classification.

Round 2

Reviewer 3 Report

Comments and Suggestions for Authors

The author has made improvements and can now be accepted